# Perturbed fatty-acid metabolism is linked to localized chromatin hyperacetylation, increased stress-response gene expression and resistance to oxidative stress

Jarmila Princová[1☯], Clàudia Salat-Canela[2☯], Petr Daněk[1], Anna Marešová[1], Laura de Cubas[2], Jürg Bähler[3], José Ayté[2], Elena Hidalgo[2], Martin Převorovský[1]*

1 Laboratory of Microbial Genomics, Department of Cell Biology, Faculty of Science, Charles University, Prague, Czech Republic, 2 Oxidative Stress and Cell Cycle Group, Universitat Pompeu Fabra, C/Dr. Aiguader, Barcelona, Spain, 3 Institute of Healthy Ageing and Department of Genetics, Evolution & Environment, University College London, London, United Kingdom

☯ These authors contributed equally to this work.
* prevorov@natur.cuni.cz

**Data Availability Statement:** The microarray data are available from the ArrayExpress database (https://www.ebi.ac.uk/arrayexpress/) under

## Abstract

Oxidative stress is associated with cardiovascular and neurodegenerative diseases, diabetes, cancer, psychiatric disorders and aging. In order to counteract, eliminate and/or adapt to the sources of stress, cells possess elaborate stress-response mechanisms, which also operate at the level of regulating transcription. Interestingly, it is becoming apparent that the metabolic state of the cell and certain metabolites can directly control the epigenetic information and gene expression. In the fission yeast *Schizosaccharomyces pombe*, the conserved Sty1 stress-activated protein kinase cascade is the main pathway responding to most types of stresses, and regulates the transcription of hundreds of genes via the Atf1 transcription factor. Here we report that fission yeast cells defective in fatty acid synthesis (*cbf11*, *mga2* and ACC/*cut6* mutants; FAS inhibition) show increased expression of a subset of stress-response genes. This altered gene expression depends on Sty1-Atf1, the Pap1 transcription factor, and the Gcn5 and Mst1 histone acetyltransferases, is associated with increased acetylation of histone H3 at lysine 9 in the corresponding gene promoters, and results in increased cellular resistance to oxidative stress. We propose that changes in lipid metabolism can regulate the chromatin and transcription of specific stress-response genes, which in turn might help cells to maintain redox homeostasis.

## Author summary

The production of fatty acids and lipids in general creates energy reserves and provides essential building blocks for cellular membranes. Oxidative stress, on the other hand, is a condition caused by increased concentration of oxidants, such as reactive oxygen species, which becomes harmful to the cells and triggers an oxidative stress response to mitigate damage. While these two processes are seemingly unrelated, we now provide evidence

accession number E-MTAB-6761. The raw ChIP-seq data are available from the ArrayExpress database under the accession number E-MTAB-11081. The scripts used for ChIP-seq data processing and analysis are available from https://github.com/mprevorovsky/ox-stress_histones.

**Funding:** This work was supported by the Univerzita Karlova v Praze [grant number PRIMUS/MED/26 to M.P.], Grantová Agentura, Univerzita Karlova [grant number GA UK 1170217 to J.P.], Ministerio de Ciencia, Innovación y Universidades (Spain) [grant numbers PGC2018-093920-B-I00 to E.H. and PGC2018-097248-B-I00 to J.A.] and Unidad de Excelencia María de Maeztu (Spain) [grant number CEX2018-000792-M to E.H. and J.A.], and a Wellcome Trust Senior Investigator Award [grant number 095598/Z/11/Z] to J.B. The funders had no role in study design, data collection and analysis, decision to publish, or preparation of the manuscript.

**Competing interests:** The authors declare that they have no conflict of interest.

that there is actually a specific regulatory connection between fatty acid metabolism and cellular resistance to oxidative stress. Using the fission *Schizosaccharomyces pombe* as a model, we show that multiple conditions that lower fatty acid production, including mutations of lipogenic enzymes and their regulators, or chemical inhibition of fatty acid synthesis, specifically boost the expression of a subset of stress-responsive genes, and increase cellular resistance to hydrogen peroxide. This regulatory link relies on histone acetyltransferases and is connected with promoter hyperacetylation at the affected genes. Our findings highlight the intricate interconnections between the metabolic state of the cell and the regulation of gene expression.

## Introduction

Oxidative stress occurs when the equilibrium between the production and the detoxification of oxidants is disturbed, leading to damage of cellular molecules [1]. Importantly, oxidative stress is associated with multiple cardiovascular and neurodegenerative diseases, diabetes, cancer, psychiatric disorders and aging [2–5]. However, under physiological conditions that create increased oxidant levels, such as mitochondrial respiration or fatty acid (FA) oxidation, the cellular antioxidant mechanisms are typically able to maintain cellular redox homeostasis and prevent oxidative stress [1,6].

In the fission yeast *Schizosaccharomyces pombe*, the Sty1 stress-activated protein kinase (SAPK) cascade, homologous to the mammalian p38 mitogen-activated protein kinase, is the main pathway responding to most types of stress conditions, including oxidative stress. Once activated, Sty1 translocates to the nucleus, where it phosphorylates and activates its main target, the basic zipper-containing transcription factor Atf1, leading to an extensive transcriptional response [7,8]. Analyses of genes showing differential expression in response to various stresses identified the core environmental stress response (CESR) as a group of genes that are jointly regulated under all or most environmental stresses. Remarkably, the regulation of most CESR genes depends on Sty1 and, to a lesser extent, on Atf1 [9,10]. During oxidative stress, the Pap1 (pombe AP-1-like) transcription factor is also involved in triggering stress gene expression, especially under less severe insults (e.g., treatment with 0.2 mM hydrogen peroxide) which are insufficient to fully activate the Sty1-Atf1 pathway. Additionally, there is crosstalk between the two pathways and some genes, such as the catalase ctt1, are regulated by both Atf1 and Pap1 [11,12]. Importantly, even under favourable conditions some level of Sty1 activity is needed for proper cell-cycle progression, especially for timing the entry into mitosis [13,14].

The CSL (<u>C</u>BF1, <u>Su</u>(H), <u>Lag</u>-1) family protein Cbf11 and the IPT/TIG ankyrin repeat-containing protein Mga2 are transcription factors regulating lipid-metabolism genes. Their target genes include the acetyl-CoA carboxylase *cut6*, the acyl-CoA desaturase *ole1*, the long chain fatty acid-CoA ligases *lcf1* and *lcf2*, or the triacylglycerol lipases *ptl1* and *ptl2* [15,16]. The loss of Mga2 causes a general disruption of the lipidome [16], while cells lacking Cbf11 have a decreased amount of lipid droplets and show mitotic defects [17]. Curiously, we have previously shown that many CESR genes are upregulated in the *cbf11Δ* deletion mutant cells, but the reason for these changes is not clear [15].

It has become apparent in recent years that the cellular metabolic state can directly affect the regulation of gene expression through the availability of selected metabolites that serve as substrates for various chromatin modifying enzymes. For example, the metabolite acetyl-CoA is central to multiple biosynthetic pathways as well as to histone acetylation by histone acetyltransferases (HATs). Perturbations in acetyl-CoA levels lead to altered histone acetylation and

gene transcription [18–20]. Acetyl-CoA is also utilized during FA synthesis, including its first and rate-limiting step catalyzed by the acetyl-CoA carboxylase (ACC). Intriguingly, ACC inhibition increases the acetylation of bulk histones and affects gene expression in yeast [21]. However, the significance and the physiological consequences of such interconnections between the metabolic state and gene expression patterns are only beginning to be understood. In this study, we show that a decrease in FA synthesis leads to increased expression of specific stress-response genes accompanied by promoter histone hyperacetylation, and to increased resistance to hydrogen peroxide ($H_2O_2$)-induced oxidative stress in fission yeast.

## Materials and methods

### Plasmid construction

The Cas9/sgRNA_TEFp (pMP134) and Cas9/sgRNA_*cbf11* (pMP153) plasmids were constructed as previously described [22]. sgRNAs targeted to the TEF promoter region of the *natMX6* cassette and to *cbf11* ORF, respectively, were inserted into the pMZ374 plasmid carrying an empty sgRNA site and a sequence encoding the Cas9 endonuclease [23]. Briefly, the whole pMZ374 was amplified by NEB Q5 polymerase (using AJ11 and AJ12, and AJ29 and AJ30 oligonucleotides, respectively), 5' ends of purified PCR products were phosphorylated, and plasmid ends were ligated together. The final plasmids were verified by restriction cleavage and sequencing. pMZ374 was a gift from Mikel Zaratiegui (Addgene plasmid # 59896; http://n2t.net/addgene:59896; RRID:Addgene_59896). Lists of oligonucleotides and plasmids used in this study are provided in Tables A and B in S1 Text, respectively.

### Strains, media and cultivations

Fission yeast cells were grown according to standard procedures [24] in either complex yeast extract medium with supplements (YES) or Edinburgh minimal medium (EMM). A list of strains used in this study is provided in Table C in S1 Text.

For construction of Cbf11-TAP scarless knock-in strain, the CRISPR/Cas9-based strategy was adapted from [22]. MP15 cells (*h- cbf11-ctap4::natR ura4-D18 leu1-32 ade6-M216*) [15] were synchronized in G1 and transformed with a fragment of the *cbf11-TAP* sequence (plasmid pMaP27 digested by SalI and EcoO109I) as template for homologous recombination together with the Cas9/sgRNA_*TEFp* plasmid (pMP134). After selection on EMM+ade+leu plates, the smallest colonies were re-streaked onto non-selective YES plates to allow for elimination of the deleterious Cas9 plasmid. The integration of *cbf11-TAP* was verified by PCR (primers MaP169 and MP28) and sequencing. Expression of Cbf11-TAP protein was verified by western blot with an anti-TAP antibody (Thermo Scientific, CAB1001). Prototrophic *cbf11-TAP* strain was then prepared by standard crossing and revalidated.

The Cbf11DBM-TAP scarless knock-in strain was constructed and validated analogously in two steps. First, to insert the DNA-binding mutation (R318H; DBM) [25] into the *cbf11* endogenous locus MaP70 cells (*h- cbf11-3HA::natMX6 ura4-D18 leu1-32 ade6-M216*) were synchronized in G1 phase and transformed with a Cas9/sgRNA plasmid targeting Cas9 next to the desired DBM mutation site in *cbf11* ORF (Cas9/sgRNA_*cbf11;* pMP153), and a *cbf11DBM-TAP* DNA fragment as template for homologous recombination (plasmid pMaP11 digested by SalI and EcoO109I). Introduction of the DBM mutation was verified by PCR (primers MP53 and MP54) coupled with restriction digestion with HpaII, and by sequencing. The resulting strain (MP670) contained the DBM mutation, but retained the HA tag and *natMX6* cassette at the *cbf11* locus. In the second step the MP670 cells were transformed with the pMP134 plasmid targeting the *natMX6* cassette, and the pMaP11 fragment described above as template for homologous recombination. The final prototrophic strain

*cbf11DBM-TAP* (MP712) was then prepared by standard crossing and validated by PCR, restriction cleavage, sequencing and western blot as described above.

All other strains constructed in this study were created using standard genetic methods [26], or the pClone system [27] for deletion of *cbf11*, *ssp2* or *mga2* genes.

### Spot tests

Exponentially growing cells were 10-fold serially diluted and spotted onto YES plates containing various concentrations of $H_2O_2$ (Sigma-Aldrich, H1009) or 125 μM menadione sodium bisulfite (Sigma-Aldrich, M5750) for oxidative stress-resistance assays, or 3-MB-PP1 (Sigma-Aldrich, 529582) for Sty1 inhibition. The spots were allowed to dry and plates were incubated at 30˚C or 32˚C until cell growth was evident. Due to the unstable nature of $H_2O_2$, the plates were poured on the day of spotting and YES agar was cooled to 45˚C prior to adding the stressor.

### RT-qPCR

The RT-qPCR results shown in all relevant figures except for Fig 1B (top panel) and 1F were obtained as follows: Total RNA was extracted from cells using the MasterPure Yeast RNA Purification Kit including a DNase treatment step (Epicentre), and converted to cDNA using random primers and the RevertAid Reverse Transcriptase kit (ThermoFisher Scientific). Quantitative PCR was performed using the 5x HOT FIREPol EvaGreen qPCR Supermix (Solis Biodyne) and the LightCycler 480 II instrument (Roche). For RT-qPCR, *act1* (actin) and *rho1* (Rho1 GTPase) were used as reference genes. The primers used are listed in Table A in S1 Text.

The RT-qPCR results shown in Fig 1B (top panel) and 1F were obtained as follows: Total RNA was extracted from 40 ml of cells at logarithmic phase by standard phenol/chloroform method, as described earlier [28]. 100 μg of total RNA was incubated with recombinant DNase I (Roche, 04716728001) for 30 min at 37˚C and the reaction was stopped by incubation at 75˚C for 10 min. Reverse transcription was performed on 2 μg of DNase-treated RNA using High-Capacity cDNA Reverse Transcription Kit (Applied Biosystems, 00777852) following manufacturer's instructions. The cDNA was diluted 1:2 prior to PCR amplification. cDNA was quantified by Real-Time PCR on Light Cycler II using Light Cycler 480 SYBR Green I Master (Roche, 04887352001).

One-sided Mann-Whitney U test was used to determine statistical significance. All statistical tests were performed on data normalized only to reference genes, however, gene expression values normalized to WT or other suitable control sample were used for plotting to allow for more intelligible visualization.

### *In vivo* measurement of roGFP2-Tpx1.C169S oxidation

Wild-type (WT) and *cbf11Δ* cells were transformed with plasmid p407.C169S, constitutively expressing the $H_2O_2$ reporter protein roGFP2-Tpx1.C169S. To measure basal and induced $H_2O_2$ levels, fluorescence of the probe was determined as described before [29]. Briefly, roGFP2 exhibits two excitation maxima at 400 nm and 475–490 nm when fluorescence emission is monitored at 510 nm, and the ratio between the two maxima varies upon oxidation by peroxides. Strains were grown in EMM to an $OD_{600}$ of 1, and fluorescence of the cultures was analyzed in 96-well plates before and after the addition of extracellular $H_2O_2$. For calculation of the degree of oxidation of the sensor (OxD), we first subtracted the equivalent fluorescence values of cells lacking the plasmid, and then used the formula displayed in [29].

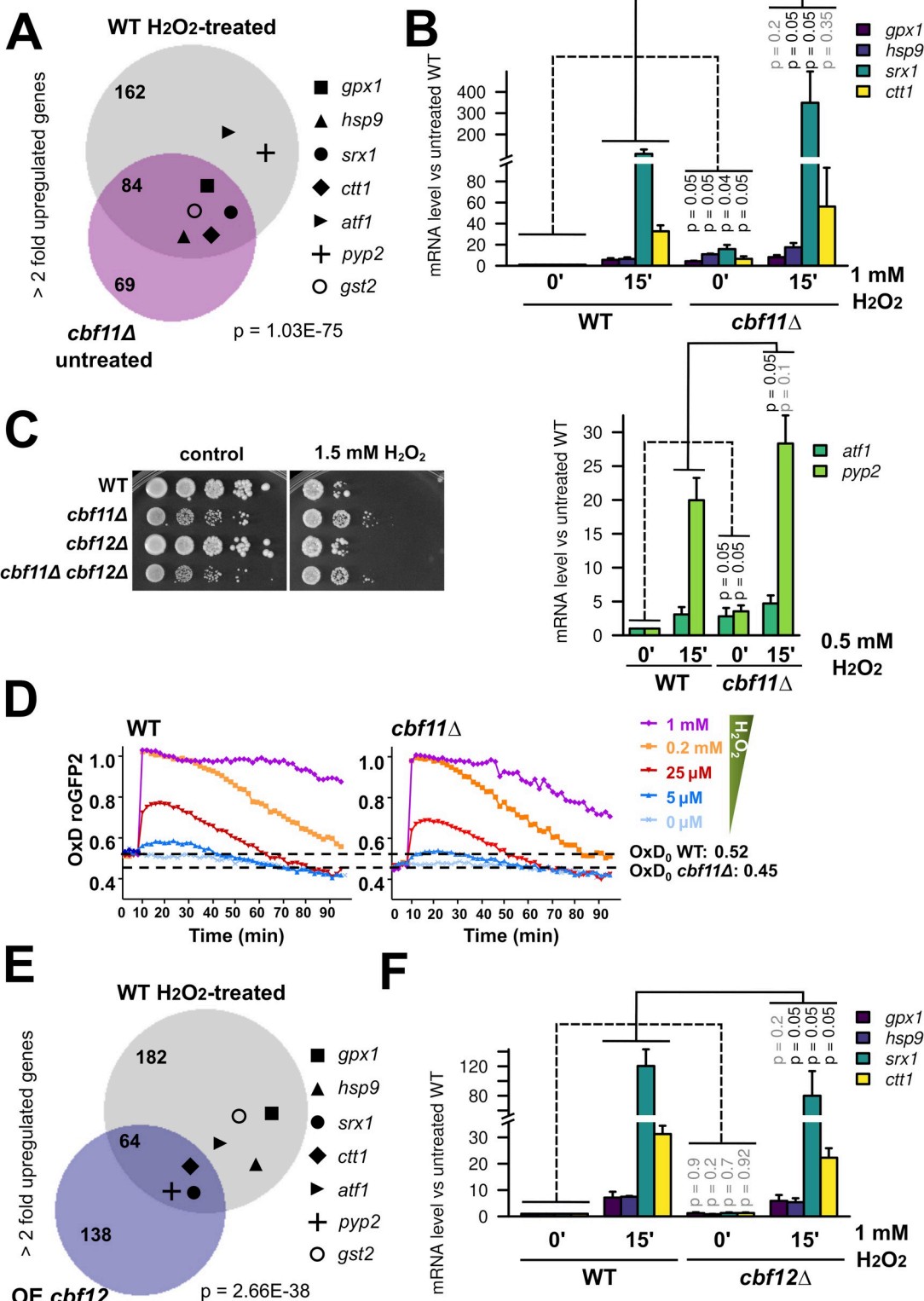

**Fig 1. Absence of Cbf11 leads to activation of stress-response genes. (A)** Venn diagram of genes upregulated more than 2-fold in WT cells after 15 min treatment with 1 mM $H_2O_2$ in EMM medium (data from [34]), and genes upregulated more than 2-fold in *cbf11Δ* cells growing exponentially in YES medium (data from [15]). The group membership of stress genes from panel B is indicated with symbols. Overlap significance was determined by two-sided Fisher's exact test. **(B)** Expression of the indicated stress genes in WT and *cbf11Δ* cells treated or not with the indicated concentrations of $H_2O_2$ for 15 min in YES was analyzed by

RT-qPCR. Mean and SD values of three independent replicates are shown. One-sided Mann-Whitney U test was used to determine statistical significance. **(C)** Survival and growth under oxidative stress of WT, *cbf11Δ*, *cbf12Δ* and *cbf11Δ cbf12Δ* cultures spotted on YES plates containing 1.5 mM $H_2O_2$. **(D)** *cbf11Δ* displays lower steady-state intracellular $H_2O_2$ levels, and detoxifies extracellular peroxides faster than WT cells. The indicated concentrations of $H_2O_2$ were directly added to EMM cultures of WT and *cbf11Δ* cells transformed with plasmid p407.C169S. The degree of probe oxidation on scale from 0 (fully reduced) to 1 (fully oxidized) is indicated on the Y axis (OxD roGFP2). The starting levels of probe oxidation in each strain background ($OxD_0$) are indicated by dashed horizontal lines. Mean values from three biological replicates are shown. **(E)** Venn diagram of genes upregulated more than 2-fold in WT cells after 15 min treatment with 1 mM $H_2O_2$ in EMM medium (data from [34]), and genes upregulated more than 2-fold in cells overexpressing Cbf12 grown in EMM (data from [15]). The group membership of stress genes from panel B is indicated with symbols. Overlap significance was determined by two-sided Fisher's exact test. **(F)** Expression of the indicated stress genes in WT and *cbf12Δ* cells treated or not with 1 mM $H_2O_2$ for 15 min in YES was analyzed by RT-qPCR. Mean and SD values of three independent replicates are shown. One-sided Mann-Whitney U test was used to determine statistical significance.

### TCA extracts and immunoblotting

TCA extracts were prepared as described [30]. Atf1 was detected with a polyclonal anti-Atf1 antibody [31], phosphorylated Sty1 was detected with an anti-phospho-p38 antibody (Cell Signaling, 9215), and total Sty1 protein was detected with a polyclonal antibody [32].

### Pap1 microscopy

Cells expressing Pap1-GFP from its endogenous chromosomal locus [33] were grown in YES to exponential phase, treated as required and fixed by 10% formaldehyde for 15 min. Then, cells were washed three times with PBS. To observe Pap1-GFP localization, cell suspension was loaded onto lectin-coated slides and imaged using the Olympus CellR microscope system.

### Microarray analysis

Cells were grown to exponential phase ($OD_{600}$ 0.5) in the YES medium at 32°C. At time 0, $H_2O_2$ was added to all cultures to the final concentration of 0.74 mM. Culture aliquots were harvested immediately before, and 15 and 60 minutes after $H_2O_2$ addition by centrifugation for 2 minutes at 1000 g, room temperature, and then snap frozen in liquid nitrogen. RNA extraction and labelling, sample hybridization to custom in-house dual-colour cDNA microarrays, and microarray data processing were performed as described previously [15]. Individual Cy3-labelled samples were hybridized together with a Cy5-labelled reference pool (an equimolar pool from all 9 samples in this study). The microarray data are available from the ArrayExpress database (https://www.ebi.ac.uk/arrayexpress/) under accession number E-MTAB-6761.

### ChIP-qPCR

ChIP-qPCR analysis of Cbf11-TAP shown in the main text was performed as described previously [15].

For ChIP-qPCR of Atf1-HA shown in the main text and ChIP-qPCR of Cbf11-TAP and Cbf11-HA shown in the supplementary files cells were grown in EMM medium and chromatin isolation was performed as described elsewhere [34]. Commercial IgG Sepharose Beads (GE Healthcare, 17-0969-01) were used in order to immunoprecipitate TAP-tagged proteins.

### H3 and H3K9ac ChIP-seq

Two independent replicates were performed. Cells were cultivated to exponential growth phase ($OD_{600}$ 0.5) in the complex YES medium and fixed by adding formaldehyde to the final concentration of 1%. After 30 min incubation, the remaining formaldehyde was quenched by 125 mM glycine. Cells were washed with PBS and broken with glass beads. Extracted

chromatin was sheared with the Bioruptor sonicator (Diagenode) using 15 or 30 cycles (for biological replicate 1 and 2, respectively) of 30 s on, 30 s off at high power settings. For all immunoprecipitations (IP) within a biological replicate the same amount of chromatin extract was used (2.5 or 3.7 mg of total protein); 1/10 of the total chromatin extract amount was kept for input DNA control. For each IP 5 µg of antibody (H3: Ab1791, H3K9ac: Ab4441, all Abcam) were incubated with the chromatin extract for 1 hour at 4˚C with rotation. Then, 50 µl of BSA-blocked Protein A-coated magnetic beads (ThermoFisherScientific, 10002D) were added to the chromatin extract-antibody suspension and incubated for further 4 hours at 4˚C with rotation. The precipitated material and input chromatin extract were decrosslinked, treated with RNase A and proteinase K. DNA was purified using phenol-chloroform extraction and sodium acetate/ethanol precipitation. In biological replicate 2, DNA purification on AMPure XP beads (Beckman Coulter, AC63880) was performed following the phenol-chloroform extraction to remove low-molecular fragments and RNA. Concentration of DNA was measured using the Quantus fluorometer (Promega) and fragment size distribution was checked on Agilent Bioanalyzer using the High Sensitivity DNA Assay. Library construction and sequencing were performed by BGI Tech Solutions (Hong Kong) using the BGISEQ-500 sequencing system.

## ChIP-seq data analysis

The *S. pombe* reference genome sequence and annotation were obtained from PomBase (release date 2018-09-04) [35,36]. Read quality was checked using FastQC version 0.11.8 (https://www.bioinformatics.babraham.ac.uk/projects/fastqc/), and reads were aligned to the *S. pombe* genome using HISAT2 2.1.0 [37] and SAMtools 1.9 [38,39]. Read coverage tracks (i.e., target protein occupancy) were then computed and normalized to the respective mapped library sizes using deepTools 3.3.1 [40]. The list of genes upregulated as part of the core environmental stress response ("CESR-UP" genes) was obtained from [9]. The CESR-UP genes were further divided into two groups based on whether or not the genes were also upregulated in untreated *cbf11Δ* cells [15]. The deepTools 3.5.1 were then used to create average-gene H3 and H3K9ac occupancy profiles for the respective CESR-UP gene subgroups and for all fission yeast genes as a control. The raw ChIP-seq data are available from the ArrayExpress database under the accession number E-MTAB-11081. The scripts used for ChIP-seq data processing and analysis are available from https://github.com/mprevorovsky/ox-stress_histones.

## Acetyl-CoA measurement

Cells grown to exponential phase (OD$_{600}$ 0.5) in the YES medium were collected by centrifugation (1000 g, 5 min) or vacuum filtration (25 ODs). The filter with cell pellet was immediately transferred to 25 ml methanol (-20˚C). Samples were centrifuged (3000 g, 5 min, -4˚C, brake 5) and supernatant was decanted. Then 410 µl of 50% freezer-cooled methanol with 10 µM PIPES was added. Cells were broken using glass beads on FastPrep 6.5 m/s, 20 s, 6 cycles. Crude extracts were ultrafiltered using 10kDa filter Amicon Ultra 0.5 ml Ultracel-10K (UFC501096). Samples were evaporated at room temperature on SpeedVac and dissolved in 40 µl 50% acetonitrile. The samples were analyzed on a Dionex Ultimate 3000RS liquid chromatography system coupled to a TSQ Quantiva mass spectrometer (ThermoScientific). A ZIC- HILIC column (150 mm × 2.1 mm, 5 µm, Merck) was used for separation of analytes. The column was maintained at room temperature and an injection of 1–2 µl of the sample was applied. The gradient elution took 20.5 min and was set from 5% A to 70% A and then 70% A was held for 2 min. A column equilibration step followed and lasted 9 min (A: 10 mM ammonium bicarbonate pH 9.3, B: 97% acetonitrile, flow rate 200 µl/min). Electrospray ionization

with switching polarity mode ran under following conditions: ion transfer tube temperature 350°C, vaporizer temperature 275°C, spray voltage 3500/3000 V (depends on the polarity mode), sheath gas 35 and aux gas 15. For targeted determination of analytes, SRM assay was developed previously by infusing pure compounds.

### Growth curves

$OD_{600}$ was recorded during 24–30 h for cells growing in YE at 30°C from an initial $OD_{600}$ of 0.1 using an automated measurement as previously described [41]. When indicated, $H_2O_2$ was added to the cultures.

## Results

### Absence of Cbf11 leads to upregulation of stress-response genes and increased resistance to $H_2O_2$

In our previous studies, we identified fission yeast genes that change their expression upon genetic manipulation of the *cbf11* and/or *cbf12* CSL transcription factor genes (deletion, over-expression). We also noted that these deregulated genes were enriched for stress-response genes [15]. More recently, we described the transcriptional signatures of wild-type (WT) cells under oxidative stress [34]. Interestingly, when comparing these datasets, we found that 55% of the genes upregulated more than two-fold in cells lacking Cbf11 were also upregulated under oxidative stress in a WT strain (p = 1.03 x $10^{-75}$; Fig 1A). This finding raises the possibility that a genuine oxidative-stress response is triggered in cells lacking Cbf11, and/or that Cbf11 acts as a direct or indirect repressor of oxidative stress-response genes.

To examine these possibilities further, we first validated our genome-wide data using RT-qPCR. To this end, we selected representative stress-response genes: *srx1*, *ctt1*, *gpx1*, *hsp9* and *gst2*, which code for sulfiredoxin, catalase, glutathione peroxidase, heat shock protein 9, and glutathione S-transferase, respectively [35]. We also included *atf1* and *pyp2*, which encode a transcription factor and a protein phosphatase that regulate the cellular response to oxidative stress [8,42], and were previously reported to be upregulated ~1.8-fold in *cbf11Δ* [15]. We found that all selected genes indeed showed moderately increased basal expression in untreated *cbf11Δ* compared to WT, and became further upregulated upon $H_2O_2$ treatment, reaching even higher transcript levels than in WT (Figs 1B and 2D).

Next, we tested whether the upregulation of oxidative stress-response genes in *cbf11Δ* cells has any physiological consequences. We found that compared to WT, *cbf11Δ* cells were more resistant to $H_2O_2$, both when grown on solid media (Fig 1C) and in liquid cultures (S1A–S1D Fig). Furthermore, when we introduced the roGFP2-Tpx1.C169S peroxide-sensitive redox probe [29], *cbf11Δ* cells showed lower basal levels of probe oxidation ($OxD_0$ of 0.45), which indicates lower steady-state levels of intracellular $H_2O_2$ compared to WT ($OxD_0$ of 0.52) (Fig 1D). Of note, a decrease of the $OxD_0$ for roGFP2-Tpx1.C169S has previously been reported for cells expressing constitutively active Sty1 [29]. Furthermore, cells lacking Cbf11 were also able to detoxify extracellular peroxides faster than WT (compare the reduction slopes of WT and *cbf11Δ* cells in Fig 1D), suggesting a higher peroxide scavenging capacity of these cells. Notably, *cbf11Δ* cells are not resistant to the superoxide generator menadione (S1E Fig), which triggers a different type of stress response than $H_2O_2$ does [10,43]. Moreover, *cbf11Δ* cells are sensitive to cold stress [44], hypoxia [16] and the microtubule poison thiabendazole [45], suggesting their resistance to $H_2O_2$ is a highly specific phenomenon. Taken together, we have confirmed that oxidative stress-response genes are moderately upregulated in untreated *cbf11Δ* cells, and these cells are resistant to oxidative stress triggered by $H_2O_2$.

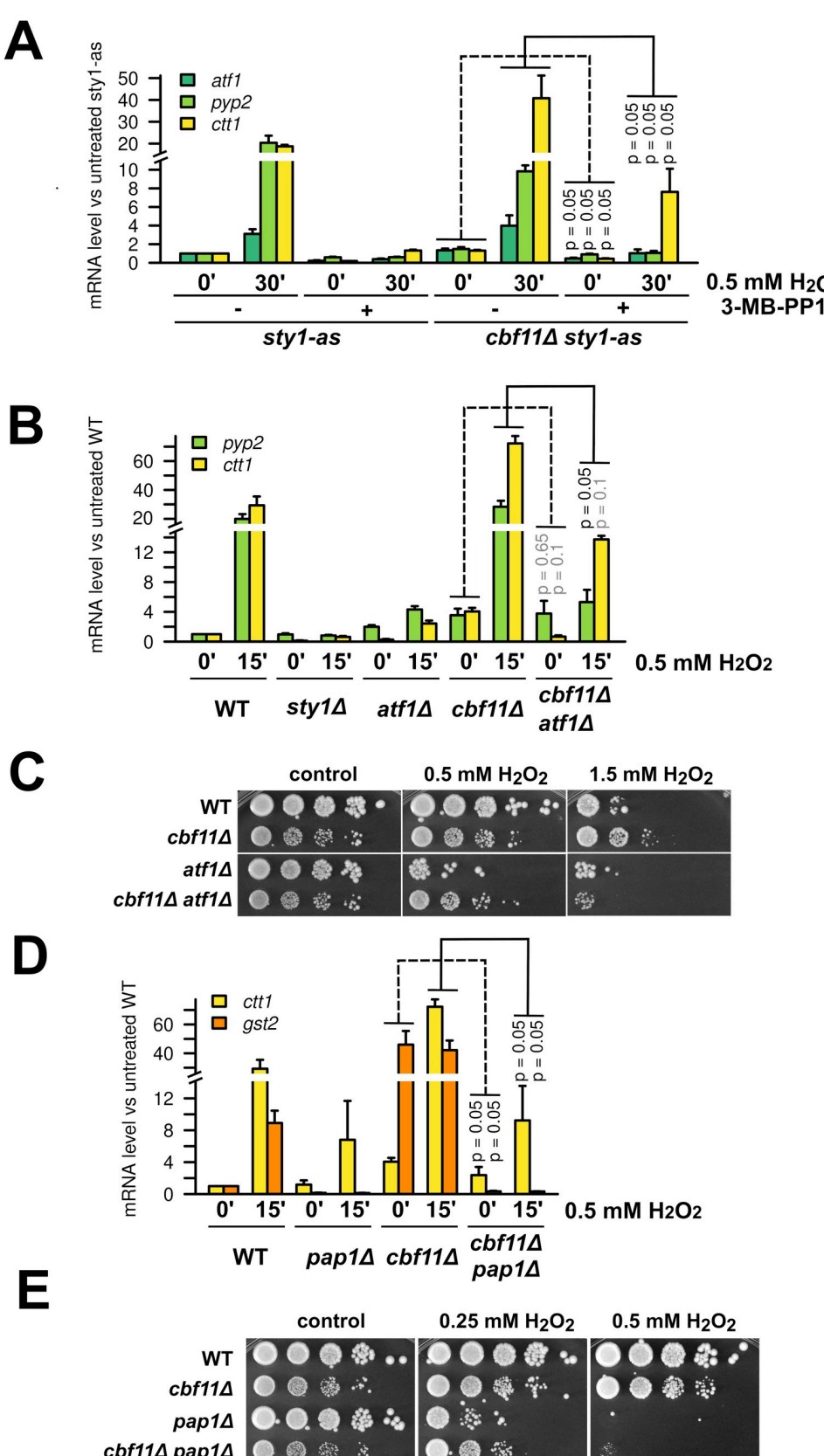

**Fig 2. -Stress-gene activation in *cbf11Δ* cells is dependent on Sty1 and Atf1. (A)** Expression of the indicated stress genes in *sty1-as* and *cbf11Δ sty1-as* cells treated or not with the Sty1-as inhibitor 3-MB-PP1 and 0.5 mM $H_2O_2$ for 30 min in YES medium was analyzed by RT-qPCR. Mean and SD values of three independent replicates are shown. One-sided Mann-Whitney U test was used to determine statistical significance. **(B)** Expression of the indicated stress genes in WT, *sty1Δ*, *atf1Δ*, *cbf11Δ*, and *cbf11Δ atf1Δ* cells treated or not with 0.5 mM $H_2O_2$ for 15 min in YES was analyzed by RT-qPCR. Mean and SD values of three independent replicates for *pyp2* and two independent replicates for *ctt1* transcript are shown. One-sided Mann-Whitney U test was used to determine statistical significance. **(C)** Survival and growth under oxidative stress of WT, *cbf11Δ*, *atf1Δ* and *cbf11Δ atf1Δ* cultures spotted on YES plates containing the indicated concentrations of $H_2O_2$. **(D)** Expression of the indicated stress genes in WT, *pap1Δ*, *cbf11Δ*, and *cbf11Δ pap1Δ* cells treated or not with 0.5 mM $H_2O_2$ for 15 min in YES was analyzed by RT-qPCR. Mean and SD values of three independent replicates are shown. One-sided Mann-Whitney U test was used to determine statistical significance. **(E)** Survival and growth under oxidative stress of WT, *cbf11Δ*, *pap1Δ* and *cbf11Δ pap1Δ* cultures spotted on YES plates containing the indicated concentrations of $H_2O_2$.

We have previously shown that Cbf12, the other fission yeast CSL paralog, acts as a Cbf11 antagonist [15,44]. Therefore, we investigated whether Cbf12 could positively regulate the transcriptional response to oxidative stress. We found that ~32% of the genes upregulated in cells overexpressing Cbf12 were also induced after oxidative stress treatment in WT cells (p = 2.66 x $10^{-38}$), even though only 3 of our selected reference genes (*ctt1*, *srx1* and *pyp2*), were upregulated more than two-fold under both conditions (Fig 1E). However, when we analyzed *cbf12Δ* cells using RT-qPCR, we did not observe any decrease in basal transcript levels of the selected stress-response genes, and we only detected a slightly weaker stress gene induction after stress imposition compared to WT (Fig 1F). Furthermore, we did not detect any notable impact of loss of Cbf12 on the transcriptome-wide response to oxidative stress (see below). Finally, the *cbf12Δ* strain did not show altered resistance to $H_2O_2$ (Fig 1C) or menadione (S1E Fig), indicating that Cbf12 only plays a minor role, if any, in the cellular response to oxidative stress. For that reason, we decided to focus on the role of Cbf11 in modulating stress-gene expression.

## Increased stress-gene expression in *cbf11Δ* cells depends on the Sty1-Atf1 and Pap1 pathways

Expression of oxidative stress-response genes and resistance to $H_2O_2$ in WT fission yeast cells are critically dependent on the Sty1 kinase and its downstream target, the Atf1 transcription factor [10,12]. Therefore, we tested the requirement for Sty1 and Atf1 in the oxidative stress-related phenotypes of *cbf11Δ* cells.

We previously failed to construct the *cbf11Δ sty1Δ* strain and noted a strong selective pressure for retaining a translocated copy of the *cbf11* gene, suggesting that the double mutant is very sick or lethal [15]. Cbf11 and Sty1 exert antagonistic effects on cell-cycle progression, where the G2/M transition is accelerated in *cbf11Δ* cells and delayed in *sty1Δ* cells. Therefore, the poor viability of *cbf11Δ sty1Δ* cells might be due to a conflict in cell-cycle regulation [15,13,14]. To circumvent these issues, we used an analog-sensitive Sty1 allele in the current study [46]. As expected, the *cbf11Δ sty1-as* double mutant showed impaired growth upon Sty1 inhibition (S2 Fig).

Next, we determined whether Sty1 kinase activity is needed for triggering the transcriptional response to $H_2O_2$ in *cbf11Δ* cells. Inhibition of Sty1-as indeed led to severely reduced induction of stress-gene transcripts in $H_2O_2$ in both WT and *cbf11Δ* (Fig 2A, timepoints 30'). Importantly, even the basal transcript levels of oxidative stress-response genes in untreated *cbf11Δ* cells were partially dependent on the Sty1 kinase activity (Fig 2A, timepoints 0'). Similarly, expression of stress-response genes in *cbf11Δ* cells was also partially dependent on Atf1 (Fig 2B), even though the impact of *atf1* deletion was not as severe as Sty1 inhibition. Furthermore, we found that Atf1 was also partially required for the increased resistance of *cbf11Δ* cells to $H_2O_2$ both on solid media and in liquid culture (Figs 2C and S1B).

Under low $H_2O_2$ concentrations, a detoxification system regulated by the Pap1 transcription factor becomes crucial to fight oxidative stress. Importantly, there is crosstalk between the Sty1 and Pap1 pathways and, furthermore, some stress-response genes are regulated by both Pap1 and Atf1 [11,12]. We therefore assessed the role of Pap1 in the increased stress-gene expression and $H_2O_2$ resistance of the *cbf11Δ* mutant. Strikingly, we found that deletion of *pap1* partially suppressed the increased expression of catalase (*ctt1*; gene coregulated by Atf1) and largely abrogated the increased expression of glutathione S-transferase 2 (*gst2*) in untreated *cbf11Δ* cells (Fig 2D). Moreover, we found that Pap1 is required for survival of *cbf11Δ* cells in the presence of low to medium doses of $H_2O_2$ (Fig 2E), which is consistent with the known critical role of Pap1 in the induction of catalase expression under oxidative stress [11,12].

In summary, these data suggest that the Sty1-Atf1 and Pap1 pathways are required for the moderate upregulation of stress-response genes observed in untreated *cbf11Δ* cells, and for the increased tolerance to $H_2O_2$ of the *cbf11Δ* strain.

## The impact of Cbf11 on stress-gene expression is likely indirect

Since the canonical stress-response pathways are required for stress-gene upregulation in *cbf11Δ* cells (Fig 2), we explored the following possibilities: 1) *cbf11Δ* cells might experience intrinsic oxidative stress that would result in the activation of the Sty1-Atf1 and/or Pap1 pathways, or 2) Cbf11 might counteract the activation of the stress-response pathways in WT cells. To test these hypotheses, we first analyzed the phosphorylation (i.e., activation) status of both Sty1 and Atf1 in cells lacking Cbf11. As shown in Fig 3A, no increase in phosphorylation of either Sty1 or Atf1 was observed in *cbf11Δ* cells, suggesting that the canonical Sty1-mediated stress response is not triggered in unstressed *cbf11Δ* cells, and neither does Cbf11 seem to block Sty1 activation.

Next, we tested whether Cbf11 could block the binding of Atf1 to stress-responsive promoters. We have previously reported two different subsets of stress genes: (i) in unstressed conditions, Atf1 is pre-bound to the first subset of genes, including *gpd1* and *hsp9*, and it is not recruited further after stress imposition; (ii) without stress, Atf1 shows relatively low occupancy at the second subset of genes, which includes *ctt1* and *srx1*, but is further recruited after stress imposition due to the activation of other transcription factors, such as Pap1 [34]. We found that Atf1 promoter occupancy in unstressed cells was not altered in *cbf11Δ* compared to WT (Fig 3B). Therefore, it is unlikely that Cbf11 could block or compete with Atf1 for binding to stress gene promoters.

Next, we tested the activation status of Pap1 in cells treated with cerulenin, an inhibitor of the fatty acid synthase (FAS) that triggers a phenotype similar to deletion of *cbf11* (see below). Pap1 undergoes activatory oxidation upon mild oxidative stress, which leads to the obstruction of its nuclear export signal. Pap1 then accumulates in the nucleus, where it facilitates stress-gene transcription [30,47,48]. Therefore, we treated WT cells expressing Pap1-GFP with cerulenin to induce *cbf11Δ*-like conditions, or with 0.2 mM $H_2O_2$, and observed Pap1 localization. While Pap1 clearly accumulated in the nucleus after the mild $H_2O_2$ treatment, Pap1 localization was indistinguishable from untreated cells after treatment with cerulenin, showing no signs of increased Pap1 activation under *cbf11Δ*-like conditions (Fig 3C).

Since Cbf11 was previously shown to co-precipitate with Atf1 [49], we also examined the possibility of Cbf11 recruitment to the promoters of stress genes, where it could directly repress their expression under non-stressed conditions, although our previous Cbf11 ChIP-seq analysis did not indicate a clear presence of Cbf11 at stress genes [15]. Nevertheless, we tried to detect Cbf11 by ChIP-qPCR at several stress-gene promoters (both in untreated and

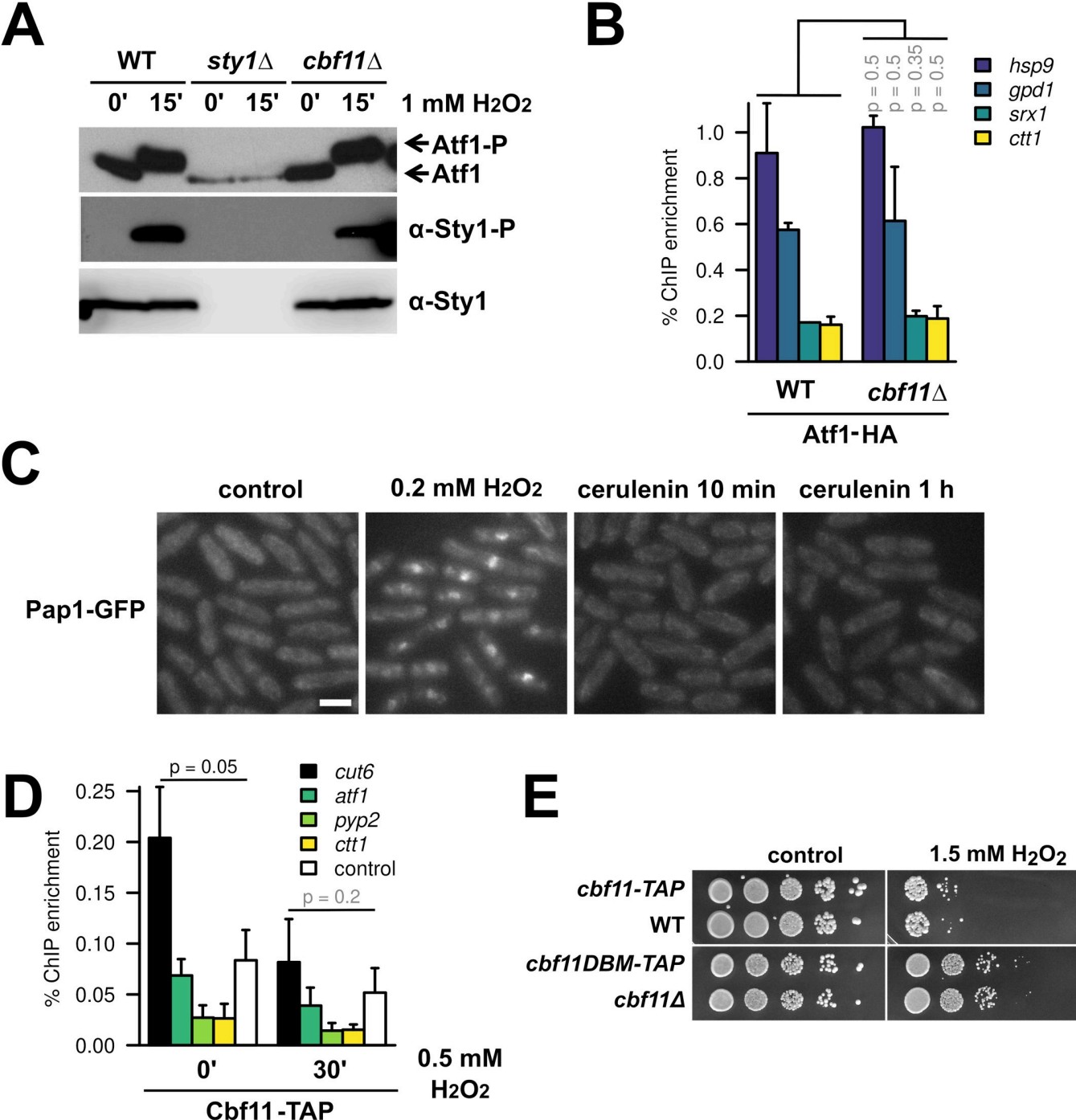

**Fig 3. Cbf11 likely affects stress-gene expression indirectly.** (A) Western blot analysis of Atf1 and Sty1 phosphorylation in WT, *sty1Δ*, and *cbf11Δ* cells treated or not with 1 mM $H_2O_2$ for 15 min in YES medium. Atf1 phosphorylation manifests as retarded migration through the gel. (B) Recruitment of Atf1 to the indicated stress-gene promoters was analyzed by ChIP-qPCR in untreated cells grown in EMM medium. Mean and SD values of two independent replicates are shown. Two-sided Mann-Whitney U test was used to determine statistical significance. (C) Pap1-GFP localization was observed in formaldehyde-fixed cells grown to exponential phase in YES using epifluorescence microscopy. Cells were either untreated (control), treated with 0.2 mM $H_2O_2$ for 25 min as a stressed control or treated with 20 μM cerulenin for 10 min and 1 hour to inhibit FA synthesis. Scale bar 5 μm. (D) Recruitment of Cbf11 to the indicated stress-gene promoters was analyzed by ChIP-qPCR in cells treated or not with 0.5 mM $H_2O_2$ for 30 min in YES. The *cut6* promoter is a positive control for Cbf11 binding [17];"control" is an intergenic locus with no Cbf11 binding (Chr I: 1,928,359–1,928,274). Mean and SD values of three independent replicates are shown. One-sided Mann-Whitney U test was used to determine statistical significance; since no stress promoter loci showed signal higher than the negative control locus, these loci were not tested statistically. (E) Survival and growth under oxidative stress of WT, *cbf11Δ*, *cbf11-TAP*, and *cbf11DBM-TAP* cultures spotted on YES plates containing 1.5 mM $H_2O_2$.

$H_2O_2$-treated cells), focusing on potential (weak) Cbf11 ChIP-seq peaks (Fig 3D) and known Atf1 binding sites (S3 Fig). The promoter of *cut6*, a well-characterized Cbf11 target gene involved in fatty-acid synthesis, served as a positive control. We assayed Cbf11 binding to DNA in both YES and EMM media, using HA- and TAP-tagged Cbf11, but we did not detect clear Cbf11 presence at the tested stress genes. Collectively, these results strongly argue that the repressive effect of Cbf11 on Sty1/Atf1-dependent stress-gene expression is brought about indirectly.

The R318H substitution in the beta-trefoil domain of Cbf11 (Cbf11DBM) abolishes its binding to the canonical CSL response element [25]. To test whether its DNA-binding activity is at all required for Cbf11 to repress Sty1-Atf1 target genes, we introduced the DBM mutation into the endogenous *cbf11* locus. Notably, the *cbf11DBM* mutant resembled the full *cbf11* deletion in that it showed increased expression of stress-response genes (Fig 4D), and it was resistant to $H_2O_2$ (Fig 3E). Thus, the DNA-binding activity is critical for the Cbf11-mediated repression of stress genes. Overall, these data suggest that Cbf11 transcriptionally regulates some other, stress-unrelated, genes that in turn affect stress-gene expression.

## Multiple lipid-metabolism mutants show derepression of stress genes and resistance to $H_2O_2$

Up to now, we analyzed the *cbf11Δ* response to oxidative stress using several representative genes. To also capture the global picture, we next performed a microarray analysis of gene expression in a timecourse experiment following $H_2O_2$ treatment. This transcriptome analysis confirmed the trends observed so far: many stress-responsive genes were moderately upregulated already in untreated *cbf11Δ* cells and were induced even further upon stress imposition (Fig 4A). Interestingly, the changes in the transcriptome of untreated *cbf11Δ* cells were mostly inductions: 134 genes were ≥2x upregulated, while only 10 genes were ≥2x downregulated compared to untreated WT cells (Fig 4B). This contrasts with the physiological reaction to $H_2O_2$ in WT cells, where stress-gene induction is accompanied by repression of numerous, mostly growth-related genes ([9] and Fig 4B, top panel). Importantly, only 2 out of the 10 genes downregulated in untreated *cbf11Δ* cells are known to be downregulated as part of the core environmental stress response in WT (*car2* and SPBPB7E8.01) [9]. These results are in agreement with our earlier notion that the increased stress-gene expression in untreated *cbf11Δ* cells is not merely a consequence of some hypothetical internal oxidative stress activating a genuine stress response.

We previously showed that several genes downregulated in *cbf11Δ* cells are related to lipid metabolism (e.g., FA synthesis) and that Cbf11 directly activates their transcription [15]. Indeed, three such direct Cbf11 target genes (*cut6*, *lcf2*, *ole1*) were also ≥2x downregulated in untreated *cbf11Δ* cells in the current microarray experiment (Fig 4C). Intriguingly, in *Saccharomyces cerevisiae* decreased FA synthesis caused by inhibition of the acetyl-CoA carboxylase (Cut6 in *S. pombe*) leads to chromatin hyperacetylation and changes in gene expression. Presumably, this is caused by increased availability of acetyl-CoA, which is used as substrate by both ACC and HATs [21]. Lipid-metabolism genes thus represent a potential link between Cbf11 transcription factor activity and changes in stress-gene expression. To examine such potential indirect regulation of stress genes by Cbf11, we assessed stress-gene expression and $H_2O_2$ resistance in other lipid-metabolism mutants. These included the Mga2 transcription factor that also regulates the ACC/*cut6* gene (*mga2Δ*; [16]) and the ACC/Cut6 enzyme itself (*Pcut6MUT* promoter mutant with ~50% reduction in expression [17] and *cut6-621* ts mutant [50]). Strikingly, all mutants showed increased expression of stress genes (Fig 4D), and all but the sick *cut6-621* ts mutant were also resistant to $H_2O_2$ (Fig 4E). To further probe the link

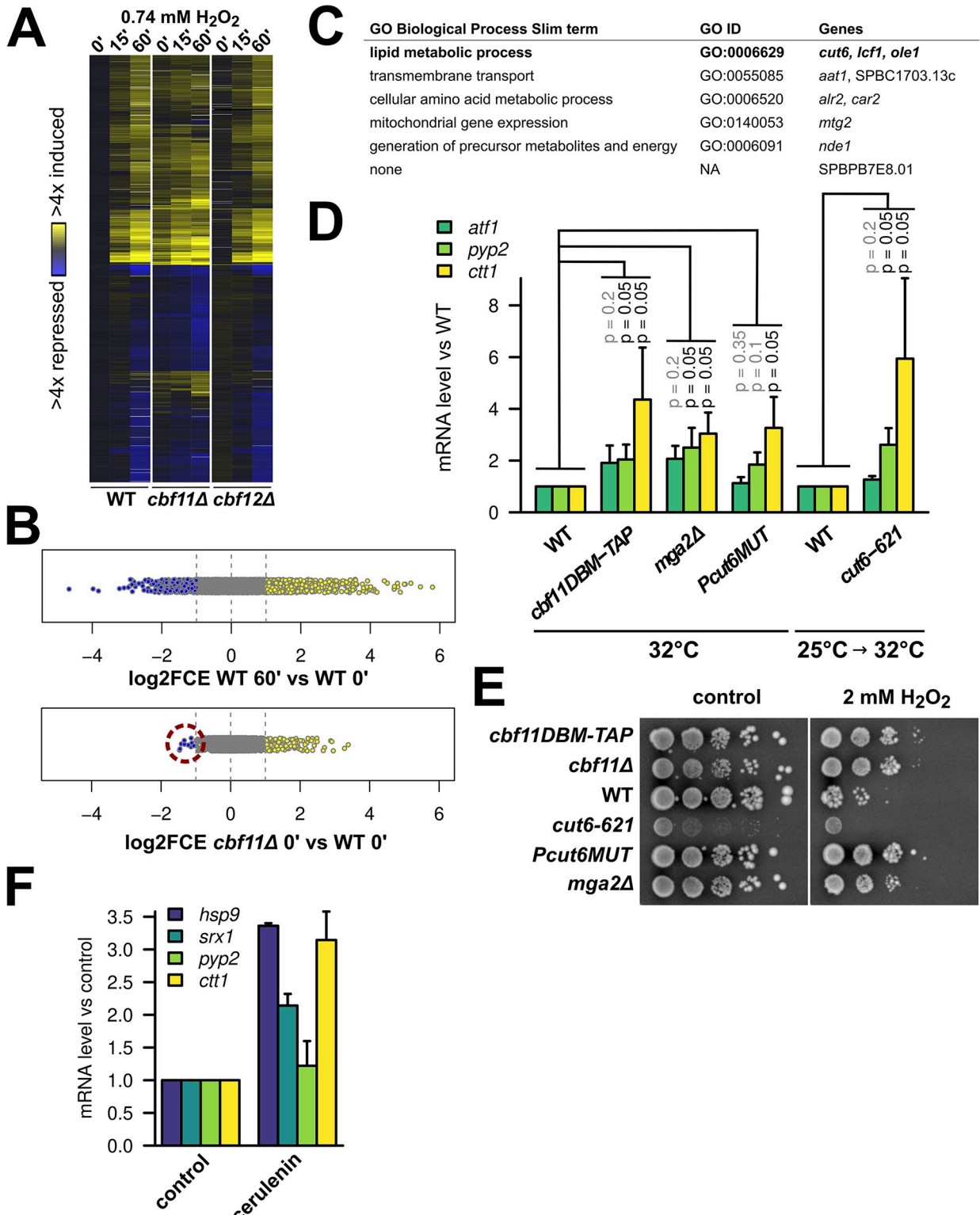

**Fig 4. Perturbation of lipid metabolism leads to derepression of stress genes.** **(A)** Heatmap of gene expression in WT, *cbf11Δ*, and *cbf12Δ* cells upon treatment with 0.74 mM $H_2O_2$ in YES medium. Transcript levels were determined by a timecourse microarray analysis and normalized to untreated WT values. Only genes showing ≥2-fold change in expression in at least one condition are shown. This experiment was performed once. **(B)** Distribution of all relative transcript levels in WT cells treated for 60 min with $H_2O_2$ and untreated *cbf11Δ* cells from the experiment in panel A. Values represent log2 of fold-changes in expression compared to untreated WT. Individual genes are shown as dots; blue and yellow dots

represent ≥2-fold down- and upregulated genes, respectively. The 10 genes downregulated in untreated *cbf11Δ* cells are highlighted with a red circle. **(C)** Biological functions of the 10 genes downregulated in untreated *cbf11Δ* cells from panel B were determined using slimmed Gene Ontology Biological Process annotations [35]. Known direct Cbf11 targets are in bold. **(D)** Expression of the indicated stress genes in untreated WT, *cbf11DBM-TAP*, *mga2Δ*, *Pcut6MUT*, and *cut6-621* cells growing in YES was analyzed by RT-qPCR. The temperature sensitive *cut6-621* mutant was shifted to a semi-restrictive temperature to inhibit Cut6 function prior to analysis. Mean and SD values of three independent replicates are shown. One-sided Mann-Whitney U test was used to determine statistical significance. **(E)** Survival and growth under oxidative stress of WT, *cbf11Δ*, *cbf11DBM-TAP*, *mga2Δ*, *Pcut6MUT*, and *cut6-621* cultures spotted on YES plates containing 2 mM $H_2O_2$. The plates were incubated at 32˚C, which is a semi-restrictive temperature for the *cut6-621* mutant. **(F)** Expression of the indicated stress genes in WT cells treated with DMSO (control) or 20 μM cerulenin for 1 hour in YES was analyzed by RT-qPCR. Mean and SD values of two independent replicates are shown.

between FA synthesis and stress-gene expression we treated exponentially growing WT cells with the FAS inhibitor cerulenin, which represents an acute intervention, as opposed to the chronic perturbations in mutants assayed so far. Also, the *fas1* and *fas2* genes, encoding the FAS subunits, are not regulated by Cbf11 [15]. Remarkably, we found that the chemical inhibition of FAS, which limits acetyl-CoA consumption by FA synthesis, also resulted in increased stress-gene expression (Fig 4F). Previously, FAS downregulation was found to increase resistance to $H_2O_2$ in the budding yeast, where the authors proposed that altered lipid composition might have lowered plasma membrane permeability for $H_2O_2$ [51]. While not being mutually exclusive, our data rather point to a novel regulatory link between FA synthesis on the one hand, and stress-gene expression and cellular resistance to oxidative stress on the other hand.

## Stress genes derepressed in *cbf11Δ* cells show H3K9 hyperacetylation at their promoters

While it was previously shown that decreased FA synthesis leads to chromatin hyperacetylation in *Saccharomyces cerevisiae*, no effect on stress-response genes was reported [21]. Intriguingly, we previously showed that the Gcn5/SAGA histone acetyltransferase regulates the expression of stress genes in fission yeast via histone H3 acetylation at lysines 9 and 14 [52,53]. We decided to explore these links further.

First, to determine whether the increased expression of stress genes in the *cbf11Δ* lipid-metabolism mutant was associated with changes in their histone acetylation profiles, we performed ChIP-seq experiments. As previously described [53], the promoters of most stress genes are largely depleted of nucleosomes (compare the H3 levels at promoters and gene bodies in Fig 5A), but we could immunoprecipitate histones both upstream and downstream of the transcription start sites (TSS in Fig 5A). Thus, we determined the occupancy of total histone H3 and H3 acetylated at lysine 9 (H3K9ac), and analyzed their distribution at stress gene bodies and promoter regions. We focused on genes upregulated as part of the core environmental stress response in WT cells (CESR-UP; [9]), and further divided these genes based on their responsiveness to the absence of Cbf11. Strikingly, the CESR-UP genes upregulated in untreated *cbf11Δ* cells (n = 94) showed a marked increase in H3K9 acetylation at their promoters and beginning of gene bodies in *cbf11Δ* cells compared to WT, while no differences in total histone H3 occupancy between the two genotypes were observed (Figs 5A and S4A). Moreover, the H3K9ac profile of the remaining CESR-UP genes (n = 441) resembled the profile of all fission yeast genes, suggesting specificity of the observed hyperacetylation (Figs 5A and S4A).

Second, we have discovered that even a modest ACC/Cut6 overexpression (~2 fold, Fig 5B) is sufficient to substantially suppress the increased stress-gene expression and oxidative-stress resistance in the *cbf11Δ* lipid-metabolism mutant (Fig 5B and 5C). Additionally, although Cut6 overexpression does not affect stress-gene expression in unstressed WT cells (Fig 5B), *cut6OE* renders WT cells sensitive to $H_2O_2$ (Fig 5C), suggesting that increased FA synthesis

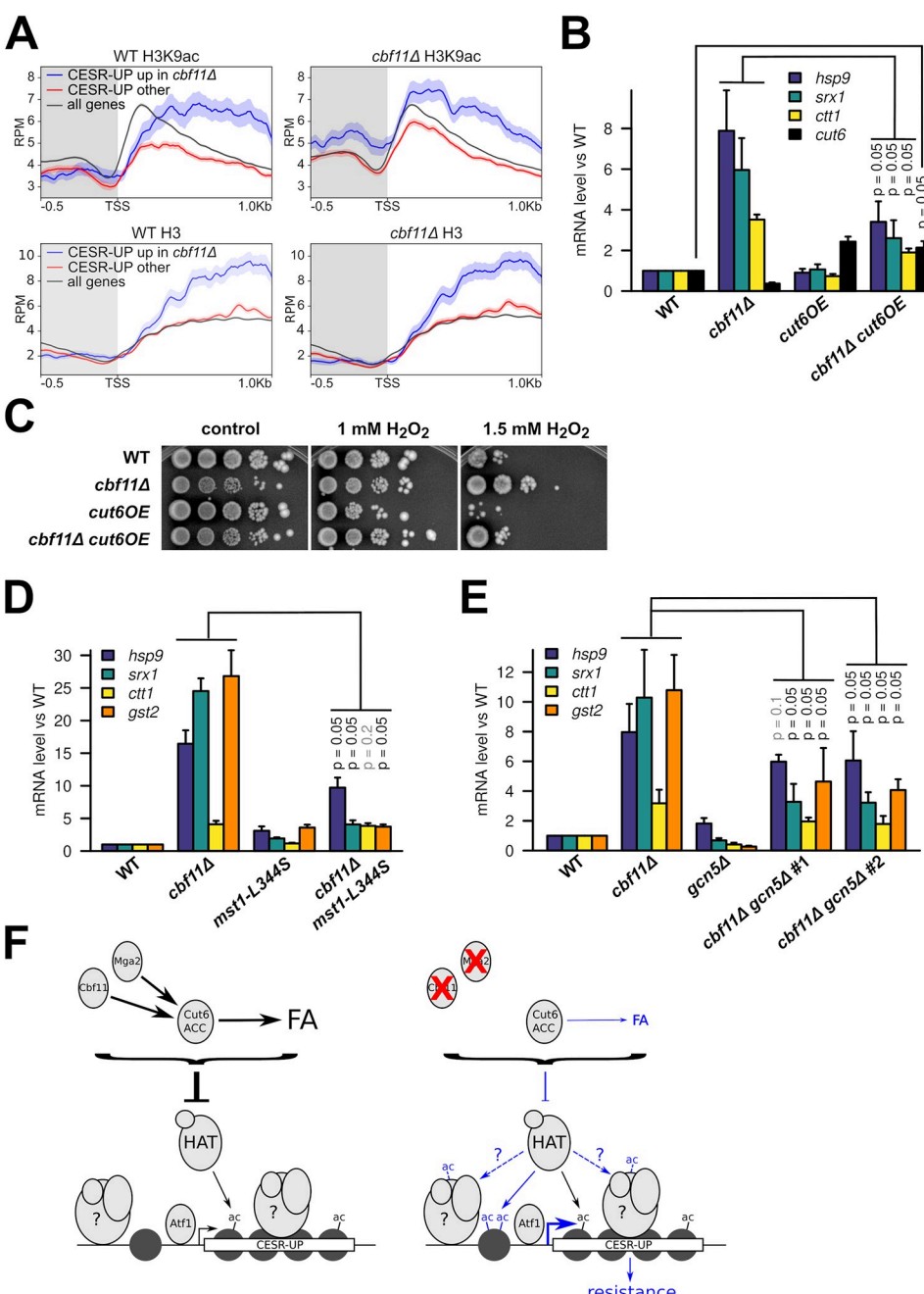

**Fig 5. Derepression of stress genes in *cbf11Δ* cells is associated with H3K9 hyperacetylation in their promoters.**
(**A**) Average gene profiles of total H3 (bottom panels) and acetylated H3K9 (top panels) occupancy at stress gene regions in WT and *cbf11Δ* cells. Genes upregulated as part of the core environmental stress response (CESR-UP, [9]) have been divided into those showing upregulation in untreated *cbf11Δ* cells (blue; n = 94, [15]) and the rest (red; n = 441). Average profile of all fission yeast genes is also shown for comparison (black; n = 6952). The curves represent mean RPM (reads per million mapped reads) values ± SEM; results from one representative biological replicate are shown. The promoter region is shaded. TSS—transcription start site. (**B**) Expression of the indicated stress genes in untreated WT, *cbf11Δ*, *cut6OE* and *cbf11Δ cut6OE* cells growing in YES medium was analyzed by RT-qPCR. Mean and SD values of three independent replicates are shown. One-sided Mann-Whitney U test was used to determine statistical significance. (**C**) Survival and growth under oxidative stress of WT, *cbf11Δ*, *cut6OE* and *cbf11Δ cut6OE* cultures spotted on YES plates containing the indicated concentrations of $H_2O_2$. (**D**) Expression of the indicated stress genes in untreated WT, *cbf11Δ*, *mst1-L344S* and *cbf11Δ mst1-L344S* cells growing in YES medium was analyzed by RT-qPCR. All cultures were shifted to 32˚C (temperature restrictive for the *mst1-L344S* strain) prior to analysis. Mean and SD values of three independent replicates are shown. One-sided Mann-Whitney U test was used to determine

statistical significance. (**E**) Expression of the indicated stress genes in untreated WT, *cbf11Δ*, *gcn5Δ*, and two independent isolates of *cbf11Δ gcn5Δ* cells growing in YES medium was analyzed by RT-qPCR. Mean and SD values of three independent replicates are shown. One-sided Mann-Whitney U test was used to determine statistical significance. (**F**) A model of the crosstalk between lipid metabolism and stress resistance. The left panel shows the WT situation; the right panel shows the situation in mutants with decreased synthesis of FA, with differences highlighted in blue. Hypothetical chromatin remodelers and/or histone modifiers are denoted with a question mark. Dashed lines represent speculative interactions. See Discussion for more details.

can hamper the launch of an effective stress response. This observation further confirms the tight relationship between FA metabolism and stress-gene expression. Since ACC activity is hypothesized to affect the general availability of acetyl-CoA [21] we also tested the global levels of acetyl-CoA in lipid metabolism mutants by liquid chromatography-mass spectrometry (LC-MS). While we detected lower acetyl-CoA levels in the *ppc1-537* phosphopantothenate-cysteine ligase mutant strain, which is severely deficient in CoA synthesis [54], we did not find any significant changes in global acetyl-CoA levels in cell extracts of *cbf11Δ* or *Pcut6MUT* (S4B Fig). These findings, however, do not completely rule out the following possibilities: 1) local, compartmentalized changes in acetyl-CoA availability occur in these two mutants [55], as the nucleocytosolic acetyl-CoA pool, which is directly affected by ACC/Cut6, is not uniform and acetyl-CoA levels at chromatin need not correlate with the global acetyl-CoA levels [56]. Nevertheless, we currently cannot explain how such precise micro-compartmentalization could be achieved; 2) Any increase in the nucleocytosolic acetyl-CoA pool is rapidly consumed by active HATs, thus leading to increased chromatin acetylation without detectable changes in the steady-state concentration of acetyl-CoA. Moreover, the *S. cerevisiae* AMPK/Snf1 (AMP-activated protein kinase) is a known inhibitor of ACC, and *snf1Δ* budding yeast cells display decreased acetyl-CoA and histone acetylation levels [57]. Curiously, the loss of the *S. pombe* AMPK ortholog *ssp2* partially suppressed the stress-gene expression in *cbf11Δ* (S5A Fig), further stressing the importance for ACC activity in regulating the stress-gene expression.

Third, we tested the importance of individual characterized *S. pombe* histone acetyltransferases for the derepression of stress genes in *cbf11Δ* cells. We also included Elp3, the elongator complex acetyltransferase, which was proposed to act only in tRNA modification [58–60], but altered histone acetylation was observed in the *elp3Δ* mutant [61]. To this end, we constructed double mutants of *cbf11Δ* and the respective HAT mutations and assayed stress-gene expression in untreated cells using RT-qPCR. We have identified the essential MYST family Mst1 (Fig 5D) and the Gcn5/SAGA acetyltransferases (Fig 5E) as dominant regulators of the stress-gene expression in lipid metabolism mutant cells, while the other tested Gcn5-related N-acetyltransferase family members Hat1 and Elp3 (S5B Fig), the MYST family protein Mst2 or H3K59-specific HAT Rtt109 (S5C Fig) were largely dispensable for stress-gene upregulation in untreated *cbf11Δ* cells.

In summary, we have demonstrated a novel regulatory link between FA metabolism and cellular resistance to oxidative stress. When FA synthesis is decreased, a subset of stress-responsive genes becomes derepressed, making cells more resistant to $H_2O_2$. Furthermore, this process is associated with increased histone acetylation at the derepressed stress genes and depends on the activity of specific HATs.

## Discussion

It is well established that metabolism and gene expression are reciprocally regulated to help cells respond efficiently to changes in both intrinsic and extrinsic factors [62]. However, the mechanisms underlying this complex regulatory crosstalk, and its diverse implications for cellular physiology are only incompletely understood. In this study, we show that perturbation of

lipid metabolism (or more specifically, the biosynthesis of FA) is associated with increased promoter histone H3K9 acetylation and HAT-dependent expression of stress-response genes, which leads to increased resistance of cells to exogenous oxidative stress. Furthermore, both the altered expression of stress genes and increased stress resistance depend on the canonical SAPK (Sty1-Atf1) and Pap1 pathways.

Both the lipid-metabolism regulators we analyzed (Cbf11, Mga2) and the Sty1 SAPK pathway affect multiple cellular processes, and their mutants show pleiotropic phenotypes. So, how can we distinguish whether our findings represent a genuine functional crosstalk between lipid metabolism and stress resistance or just some indirect effects of cellular stress? First, diverse types of stress activate a common set of stress-response genes and can lead to cross-protection against other, unrelated stresses [9,63]. Conversely, a mild dose of a particular stress can precondition cells to cope with a much higher dose of the same type of stress [64,65]. Several lines of evidence indicate that none of this can explain our observations, and that the increased resistance to oxidative stress of lipid-metabolism mutants is not merely due to the cells being intrinsically stressed. While proteins of the major stress-response pathways are indeed required for the increased oxidative-stress resistance of lipid-metabolism mutants, we have not detected increased levels of reactive oxygen species in untreated *cbf11Δ* cells (Fig 1D), and neither Sty1-Atf1 nor Pap1 were hyperactivated in untreated *cbf11Δ* cells (Fig 3A and 3C). Note that Sty1 also regulates entry into mitosis, so it is active to some extent even in unstressed cells, and only becomes hyperactivated under stressful conditions [13,14]. Moreover, our analysis of the transcriptome of untreated *cbf11Δ* cells identified mainly gene upregulation, which is unlike a typical stress response where a large group of genes is downregulated (Fig 4A and 4B). Second, the specificity of lipid-metabolism impingement on stress resistance is underlined by the phenotypes of the *cut6* acetyl-CoA carboxylase mutants (Figs 4D, 4E and 5B). In particular, the well-defined *Pcut6MUT* promoter mutation, which results in ~50% reduction of *cut6* mRNA levels and decreased amount of functional ACC/Cut6 protein without any notable pleiotropic defects [17], does also lead to increased stress-gene expression and resistance to oxidative stress, similar to *cbf11Δ* and *mga2Δ*. On the other hand, *cut6* overexpression in WT cells makes them more sensitive to oxidative stress (Fig 5C). Furthermore, the $H_2O_2$-resistant *cbf11Δ* mutant is not resistant to superoxide stress (S1E Fig), and is actually sensitive to a number of other stresses [16,44,45], highlighting the specificity of Cbf11 impact on stress gene expression. Taken together, the crosstalk between lipid metabolism and stress resistance appears to be a genuine phenomenon, and not just a side effect of a pleiotropic mutant phenotype.

How is the crosstalk between lipid metabolism and stress resistance mediated? Our results show that the DNA-binding activity of Cbf11 plays a critical role (Fig 3D), but Cbf11 does not seem to regulate stress-gene expression directly (Figs 3C and S3). Therefore, some Cbf11 (and/or Mga2) target gene(s) likely provide the connection between lipid metabolism and stress-gene expression. The Cut6 ACC is a strong candidate: 1) the *cut6* gene is regulated both by Cbf11 and Mga2 [16,17]; 2) ACC is a major consumer of acetyl-CoA, capable of affecting histone acetylation levels via limiting acetyl-CoA availability for HATs [21]; 3) hyperactive ACC results in hypoacetylated chromatin and stress-sensitive cells in budding yeast [57]; 4) decreased or increased expression of *cut6* alone results in inverse changes in the expression of stress genes (Figs 4D and 5B) and resistance to $H_2O_2$ (Figs 4E and 5C); and 5) promoters of stress genes upregulated in the *cbf11Δ* mutant showed increased H3K9 acetylation (Figs 5A and S4A). Importantly, H3K9 acetylation levels correlate with transcription [66], enhance binding of transcription factors [67] and promote transcriptional elongation by RNA polymerase II [52,68]. Gcn5/SAGA is a major H3K9-targeting HAT in the fission yeast [61], and it is important for proper stress-gene activation during oxidative stress [52]. Our data show that

Gcn5 is indeed important for the increased stress-gene expression in the *cbf11Δ* lipid mutant (Fig 5E). Strikingly, the evolutionarily conserved MYST family histone acetyltransferase Mst1 (ortholog of human Tip60, *S. cerevisiae* Esa1) with broad enzymatic specificity for both histone and non-histone targets [69,70] is also strongly required for the increased stress-gene expression in *cbf11Δ* cells (Fig 5D). Since Mst1 is not known to acetylate H3K9, some non-histone proteins, such as chromatin remodelers [71,72] or other transcription regulators [73], may be regulated by Mst1-dependent acetylation and help project the metabolic state into changes in gene expression (Fig 5F). Interestingly, *S. cerevisiae* Esa1 physically interacts with the stress-responsive transcription factors Msn2, Msn4 and Yap1 [69,74]. Thus, HAT activity and stress-gene promoter acetylation represent plausible candidates for the mechanistic link between lipid metabolism and stress-gene expression. We speculate that such changes in acetylation of histones or non-histone targets may create a more transcription-competent environment at stress-gene promoters, increasing moderately their basal (Sty1-Atf1 and/or Pap1-dependent) transcription rates in unstressed cells, and also boosting their ability to be induced during stress (Figs 1B and 2B).

While it is also formally possible that Cbf11 might directly repress the activity of Gcn5 and/ or Mst1, we consider such scenario unlikely. First, Cbf11 does not bind to stress gene promoters, where it could have potentially restricted access to HATs or repressed HAT activity (Figs 3C and S3). Second, Cbf11 does not bind to the promoters of *gcn5* and *mst1*, and *gcn5* and *mst1* transcript levels do not change in *cbf11Δ* cells [15]. Third, stress gene expression is upregulated even in the *Pcut6MUT* mutant or upon cerulenin treatment (Fig 4D and 4F), where Cbf11 is intact, so the physical presence/absence of Cbf11 seems not to be critical.

The next question then is how specificity is achieved—why and how only a subset of stress-response genes become specifically upregulated upon perturbation of FA synthesis. Notably, the subset of CESR-UP genes upregulated in untreated *cbf11Δ*, whose promoters show H3K9 hyperacetylation in *cbf11Δ* cells, tend to have above-average nucleosome occupancy in their transcribed regions, unlike the other CESR-UP genes (see the total H3 levels in Figs 5A and S4A). This suggests that a particular chromatin structure and/or presence of a specific ensemble of chromatin remodelers and histone modifying enzymes make those genes more responsive to changes in the metabolic state. Notably, Gcn5 and Mst1, the HATs required for increased stress-gene expression in *cbf11Δ* cells, are the catalytic subunits of the histone acetyltransferase modules of transcription co-activator complexes SAGA and NuA4, respectively [75]. It is conceivable that other subunits or modules of the SAGA and NuA4 complexes might be responsible for the observed specificity in lipid metabolism-regulated transcription, e.g. by directing HAT complex recruitment to particular genes or by affecting HAT complex interactions with other proteins. For instance, the SAGA subunit Tra1 mediates the interaction between specific transcription factors at some loci but is dispensable at others [76]. Overall, however, the specificity of HATs towards gene promoters is not sufficiently characterized. Intriguingly, in mammalian cells, lipid-derived acetyl-CoA can provide up to 90% of acetyl-carbon for histone acetylation, and supplementation with the octanoate FA (which is turned into acetyl-CoA by beta-oxidation) results in histone hyperacetylation and induction of specific genes, distinct from those induced by glucose-derived acetyl-CoA. However, the mechanism whereby specificity is achieved is not clear [77]. Curiously, the ACSS2 acetyl-CoA synthetase was found to be physically associated with chromatin in mouse neurons, where it affects histone acetylation and expression of specific memory-related genes by targeted, on-site production of the acetyl-CoA substrate for HATs [78]. Importantly, in *S. pombe* the Acs1 acetyl-CoA synthetase is the key contributor to the nucleocytosolic acetyl-CoA pool [79], and it localizes predominantly to the nucleus [54]. Recruitment of Acs1 to specific genes could

thus potentially provide another means for the required specificity in transforming metabolic changes into changes in gene expression.

Finally, what is the purpose of such a crosstalk between lipid metabolism and stress-gene expression? In other words, what is the selective advantage of upregulating specific stress-response genes during decreased FA synthesis? Specific metabolic states are often associated with increased levels of distinct stressors, which can perturb cellular homeostasis. A metabolic control of the stress response would allow cells to better adapt to changes in cellular chemistry and ensure that any potential damage to cellular components is minimized. For example, FA synthesis is downregulated upon carbon-source limitation [80]. This is often followed by lipolysis and increased FA oxidation to compensate for the lack of energy resources. Notably, these FA catabolic processes generate increased levels of reactive oxygen species [81]. Therefore, a timely mild upregulation of oxidative stress-response genes in response to decreased FA synthesis could represent a useful safety precaution for the cell. Conversely, oxidative stress may, intriguingly, feed back to the regulation of lipid metabolism, as ACC/*cut6* transcript levels decrease upon treatment with $H_2O_2$ [9]. Curiously, perturbations of FA metabolism have been linked to altered stress resistance in *Caenorhabditis elegans*, even though the mechanism underlying this connection was not determined [82]. Another *C. elegans* study found that the NHR-49 nuclear hormone receptor, a known transcriptional regulator of lipid-metabolism genes, can also (perhaps indirectly) regulate the transcriptional response to fasting and peroxide stress, and is required for resistance to organic peroxides [83].

Moreover, several intriguing studies have described crosstalk between FA metabolism and stress resistance in mice. First, β-hydroxybutyrate, a ketone body produced from oxidized FA during fasting, prolonged exercise or in patients with diabetes, has been reported to protect against oxidative stress in the mouse kidney. β-hydroxybutyrate inhibits class I histone deacetylases, thereby enhancing H3K9 and H3K14 promoter acetylation and transcription of several stress genes [84]. Second, elevation of acetyl-CoA levels by octanoate supplementation reduced ischemia/reperfusion injury in the heart through promoting histone acetylation and antioxidant gene expression, thus inhibiting cardiomyocyte apoptosis [85]. Moreover, inhibition of FA synthesis by knocking out ACC in the mouse liver led to increased hepatic tumor burden, likely by increasing the resistance of tumor cells to oxidative stress [86]. These studies underscore that increased cellular resistance to oxidative stress can have both beneficial and detrimental consequences, and the use of ACC inhibitors for treating cancer in humans might bring about undesired development of resistance to standard ROS-generating chemotherapy. On the other hand, even a change of lipid metabolism in the opposite direction, namely increased synthesis of (saturated) FA, can protect cancer cells from oxidative stress by making their membrane lipids less susceptible to ROS and by affecting membrane permeability for chemotherapeutics [87]. Thus, the complex situation in metazoa highlights the importance of the particular cellular and organismal context for the final outcome of the crosstalk between FA metabolism and stress resistance.

## Supporting information

**S1 Fig. Resistance of *cbf11Δ* cells to oxidative stress partially depends on Atf1 and is specific for hydrogen peroxide. (A-D)** Growth curves of stress-related mutants in the presence or absence of the indicated concentrations of $H_2O_2$ in YES medium. The *pyp1Δ* and *sty1Δ* strains represent strongly resistant and strongly sensitive controls, respectively. **(E)** Survival and growth under superoxide stress of WT, *cbf11Δ*, *cbf12Δ* and *cbf11Δ cbf12Δ* cultures spotted on YES plates containing 125 μM menadione.
(TIF)

**S2 Fig. Lack of Sty1 activity further impairs growth of *cbf11Δ* cells.** Exponentially growing WT, *cbf11Δ*, *sty1-as*, *cbf11Δ sty1-as*, and *sty1Δ* cultures were spotted on YES plates containing 10 μM Sty1-as inhibitor 3-MB-PP1 and incubated for the indicated number of days.
(TIF)

**S3 Fig. Cbf11 does not bind to Atf1 binding sites at stress-gene promoters. (A,B)** Recruitment of Cbf11-HA and Cbf11-TAP, respectively, to known Atf1 binding sites in the indicated stress gene promoters was analyzed by ChIP-qPCR in cells treated or not with 1 mM $H_2O_2$ for 5 min in EMM medium. The *cut6* promoter is a positive control for Cbf11 binding [15]; "control" is a locus with no expected Cbf11 binding. Mean and SD values of two (A) and three (B) independent replicates are shown. One-sided Mann-Whitney U test was used to determine statistical significance.
(TIF)

**S4 Fig. Derepression of stress genes in *cbf11Δ* cells is associated with H3K9 hyperacetylation in their promoters (the second replicate) but not with altered total cellular acetyl-CoA levels. (A)** Average gene profiles of total H3 (bottom panels) and acetylated H3K9 (top panels) occupancy at stress-gene regions in WT and *cbf11Δ* cells, respectively. Genes upregulated as part of the core environmental stress response (CESR-UP, [9]) have been divided into those showing upregulation in untreated *cbf11Δ* cells (blue; n = 94, [15]) and the rest (red; n = 441). Average profile of all fission yeast genes is also shown for comparison (black; n = 6952). The curves represent mean RPM (reads per million mapped reads) values ± SEM. The promoter region is shaded. TSS—transcription start site. **(B)** Total cellular acetyl-CoA levels in WT, *ppc1-537*, *cbf11Δ* and *Pcut6MUT* cell extracts were determined by LC-MS. Mean and SD values of four independent replicates for *ppc1-537* and five independent replicates for all other strains tested are shown. Two-sided Mann-Whitney U test was used to determine statistical significance.
(TIF)

**S5 Fig. AMPK and histone acetyltransferases Elp3, Hat1, Rtt109 and Mst2 have limited effect on stress gene derepression in *cbf11Δ* cells. (A,B,C)** Expression of the indicated stress genes in cells growing in YES medium was analyzed by RT-qPCR. Mean and SD values of three independent replicates are shown. One-sided Mann-Whitney U test was used to determine statistical significance.
(TIF)

**S1 Text. Table A**–List of oligonucleotides; Table B–List of plasmids; Table C–List of strains.
(DOCX)

**S1 Data. Source numerical data.**
(XLSX)

## Acknowledgments

The authors would like to thank Eishi and Chiaki Noguchi for providing the *mst1ts* strain, Anna Janovská and the Laboratory of Mass Spectrometry at Biocev Research Center, Faculty of Science, Charles University for performing the LC-MS analyses, NBRP Japan for providing the *ppc1-537* strain, Simona Veselá, Patrik Hohoš and Viacheslav Zemlianski for help with the *ssp2Δ* and *cbf11Δ ssp2Δ* strain construction, all members of the GenoMik and ReGenEx groups for their support and insightful discussions, and Eva Krellerová, Adéla Kracíková and Kateřina Svobodová for their technical assistance.

## Author Contributions

**Conceptualization:** Jarmila Princová, Jürg Bähler, José Ayté, Elena Hidalgo, Martin Převorovský.

**Data curation:** Jarmila Princová, Clàudia Salat-Canela, Martin Převorovský.

**Formal analysis:** Jarmila Princová, Clàudia Salat-Canela, Petr Daněk, Anna Marešová, Laura de Cubas, José Ayté, Elena Hidalgo, Martin Převorovský.

**Investigation:** Jarmila Princová, Clàudia Salat-Canela, Petr Daněk, Anna Marešová, Laura de Cubas, Martin Převorovský.

**Resources:** Anna Marešová, Laura de Cubas, Jürg Bähler.

**Supervision:** José Ayté, Elena Hidalgo, Martin Převorovský.

**Visualization:** Jarmila Princová, Clàudia Salat-Canela, Laura de Cubas, Martin Převorovský.

**Writing – original draft:** Jarmila Princová, Clàudia Salat-Canela, José Ayté, Elena Hidalgo, Martin Převorovský.

**Writing – review & editing:** Jarmila Princová, Anna Marešová, Jürg Bähler, José Ayté, Elena Hidalgo, Martin Převorovský.

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
