## [Decision Letter · Decision Letter 0]

25 Nov 2022

Dear Dr Převorovský,

Thank you very much for submitting your Research Article entitled 'Perturbed fatty-acid metabolism is linked to localized chromatin hyperacetylation, increased stress-response gene expression and resistance to oxidative stress' to PLOS Genetics. The reviews from Review Commons were fully considered. In addition, two reviewers were asked to evaluate your response to the Review Commons comments.

There is some disagreement between the two reviewers. Both agree that the manuscript carefully links fatty acid biosynthesis with resistance to oxidative stress. However, they both reiterate points from the first reviews, that it is very difficult to envisage how acetyl-CoA availability could be locally restricted. In the results section you describe measurements showing that global levels of acetyl-CoA do not vary, but the discussion section does not adequately discuss the ramifications of this for your model. You need to clearly discuss possible mechanisms for restricting acetyl-CoA , and also to consider alternative possibilities. Reviewer 2 has suggested one alternative - that Cbf11 directly regulates activity of Gcn5/Mst1. Reviewer 2 suggests some other experimental analyses that are not necessary for the current manuscript. 

We therefore ask you to modify the manuscript, specifically addressing the comments on acetyl-CoA availability.

Yours sincerely,

Geraldine Butler

Section Editor

PLOS Genetics

Reviewer's Responses to Questions

**Comments to the Authors:**

Reviewer #1: In the revised version, the authors present new data, and respond to the criticism. They provide evidence that Pap1 is needed for stress gene activation when lipid metabolism is perturbed. Importantly, a new experiment has been added showing that overexpression of cut6+ does make cells more sensitive to H2O2 (Fig. 5C), further demonstrating that histone acetylation is important for oxidative stress resistance.

My main concern was the authors’ claim that altered acetyl-CoA availability during oxidative stress or defects in lipid metabolism was the cause of altered histone acetylation at promoters, which in turn affects transcription.

The first problem is that the authors have measured and not found any change in global acetyl-CoA levels. They then reason that local availability of this molecule at certain promoters could still be altered.

The problem then becomes to explain how this could selectively affect certain promoters. The authors argue that even if acetyl-CoA is a small and freely diffusible molecule which should quickly redistribute within a certain cellular compartment, there is evidence for separate pools of nuclear and cytoplasmic acetyl-CoA pools. In my view, this is not sufficient evidence. We would have to postulate a mechanism for highly localized subpools around certain promoters. One way I could imagine this to happen would be by phase separation, forming liquid condensates that slow down diffusion, and I am not aware of evidence for such condensates here. One could also imagine acetyl-CoA synthesis physically linked to e.g. chromosome domains or promoters. This remains speculation however.

Nevertheless, I find that overall, there is sufficient interesting new data in the manuscript to warrant publication, even if I remain unconvinced about the proposed mechanism.

I would like to see the authors further develop their arguments about how acetyl-CoA availability could be locally restricted (to very small volumes, certainly not the whole nucleus) in the cell to explain their observations. Are there alternative possibilities?

Reviewer #2: Reviewer's comments were uploaded as an attachment.

**Have all data underlying the figures and results presented in the manuscript been provided?**

Reviewer #1: Yes

Reviewer #2: Yes

PLOS authors have the option to publish the peer review history of their article (what does this mean?). If published, this will include your full peer review and any attached files.

Reviewer #1: **Yes: **Per Sunnerhagen

Reviewer #2: No

---

## [Editor Report · Decision Letter 1]

19 Dec 2022

Dear Dr Převorovský,

We are pleased to inform you that your manuscript entitled "Perturbed fatty-acid metabolism is linked to localized chromatin hyperacetylation, increased stress-response gene expression and resistance to oxidative stress" has been editorially accepted for publication in PLOS Genetics. Congratulations!

Yours sincerely,

Geraldine Butler

Section Editor

PLOS Genetics

Geraldine Butler

Section Editor

PLOS Genetics

Comments from the reviewers (if applicable):

**Data Deposition**

http://datadryad.org/submit?journalID=pgenetics&manu=PGENETICS-D-22-01201R1

**Press Queries**

---

## [Editor Report · Acceptance letter]

5 Jan 2023

PGENETICS-D-22-01201R1 

Perturbed fatty-acid metabolism is linked to localized chromatin hyperacetylation, increased stress-response gene expression and resistance to oxidative stress 

Dear Dr Převorovský, 

We are pleased to inform you that your manuscript entitled "Perturbed fatty-acid metabolism is linked to localized chromatin hyperacetylation, increased stress-response gene expression and resistance to oxidative stress" has been formally accepted for publication in PLOS Genetics! Your manuscript is now with our production department and you will be notified of the publication date in due course.

With kind regards,

Bernadett Koltai

PLOS Genetics

On behalf of:
